# Immunohistochemical analysis of CD155 expression in triple-negative breast cancer patients

**Katsuhiro Yoshikawa**[1,2], **Mitsuaki Ishida**[1]\*, **Hirotsugu Yanai**[2], **Koji Tsuta**[1],
**Mitsugu Sekimoto**[2], **Tomoharu Sugie**[2]

1 Department of Pathology and Clinical Laboratory, Kansai Medical University, Osaka, Japan, 2 Department of Surgery, Kansai Medical University, Osaka, Japan

\* ishidamt@hirakata.kmu.ac.jp

## Abstract

### Introduction

CD155 is an immune checkpoint protein. Its overexpression is an indicator of poor prognosis in some types of cancer. However, the significance of CD155 expression in patients with triple-negative breast cancer, and the relationship between CD155 and programmed death-ligand 1 (PD-L1) expression, have not yet been analyzed in detail.

### Methods

Using immunohistochemical staining and tissue microarrays, we analyzed the expression profiles of CD155 and PD-L1 in 61 patients with triple-negative breast cancer. Relapse-free survival and overall survival rates were compared according to CD155 expression. The correlation between CD155 expression and clinicopathological factors, including PD-L1 expression (using SP142 and 73–10 assays), was also examined.

### Results

CD155 expression was noted in 25 patients (41.0%) in this cohort. CD155 expression did not correlate with pathological stage, histological grade, Ki-67 labeling index, or stromal tumor-infiltrating lymphocytes. Only PD-L1 expression in tumor cells by SP142 assay significantly correlated with CD155 expression (p = 0.035); however, PD-L1 expression in tumor cells by 73–10 assay did not show a correlation (p = 0.115). Using the 73–10 assay, 59% of patients showed CD155 and/or PD-L1 expression in tumor cells. Moreover, using the SP142 assay, 63.3% of patients showed CD155 and/or PD-L1 expression in immune cells. CD155 expression did not correlate with either relapse-free survival or overall survival (p = 0.485 and 0.843, respectively).

### Conclusions

CD155 may be a novel target for antitumor immunotherapy. The results of this study indicate that CD155 may expand the pool of candidates with triple-negative breast cancer who could benefit from antitumor immunotherapy.

**Data Availability Statement:** All relevant data are within the manuscript and its Supporting Information files.

**Funding:** This study was partially supported by a research grant D2 from Kansai Medical University (awarded to K.Y.). No additional external funding was received for this study. The funders had no role in study design, data collection and analysis, decision to publish, or preparation of the manuscript.

**Competing interests:** The authors have declared that no competing interests exist.

## Introduction

CD155, also known as nectin-like 5 (NECL5), is a type I transmembrane glycoprotein that belongs to the immunoglobulin superfamily, and is involved in many physiological processes [1–3]. This protein was initially identified as a poliovirus receptor [4], and has been shown to play important roles in cell adhesion and proliferation [1,2]. Its overexpression has been reported to promote tumor invasion and proliferation, and is associated with poor prognosis in some types of cancer, including colorectal cancer, lung adenocarcinoma, pancreatic cancer, and hepatocellular carcinoma [5–8].

Recently, CD155 has attracted considerable attention as an antitumor therapeutic target because of its immunoregulatory functions [1,9,10]. CD155 has an immunoreceptor tyrosine-based inhibition motif (ITIM) [2], and is a ligand for DNAX-associated molecule 1 (DNAM-1), T-cell immunoglobulin and ITIM domain (TIGIT), and CD96 [9,10]. Interaction with DNAM-1, which is expressed on natural killer (NK) cells and cytotoxic T-cells, leads to activation of immune reactions; in contrast, interaction with TIGIT and CD96, which are expressed on NK cells and T-cells, results in immunosuppression [1,9,10]. It has been speculated that the functions of CD155 are modified by the tumor microenvironment, leading to tumor immunosuppression [9]; therefore, CD155 has been recognized as a potential target for antitumor therapy [9].

The efficacy of anti-programmed death ligand 1 (PD-L1)/programmed cell death 1 (PD-1)-targeted therapy has been well recognized in some types of cancer, including non-small cell lung cancer and malignant melanoma [11,12]. Triple-negative breast cancer (TNBC) is the most aggressive subtype of breast cancer, which is characterized by the lack of expression of hormone receptors and human epidermal growth factor receptor 2 (HER2) [13,14]. Recent studies [15,16] have revealed that anti-PD-L1/PD-1-targeted therapy resulted in a significant improvement in the survival of patients with TNBC. However, the number of candidates for this novel therapy is small. Therefore, novel targets for cancer immunotherapy are required. The expression status of CD155, an immune checkpoint protein, in patients with TNBC is not well known [17–19]. Additionally, the association between CD155 and PD-L1 expression in patients with TNBC has not yet been analyzed. The aim of this study was to examine the immunohistochemical expression of CD155, and the correlation between its expression and clinicopathological features, as well as comparing CD155 and PD-L1 expression using SP142 and 73–10 assays, in patients with TNBC.

## Materials and methods

### Patient selection

We selected 165 consecutive patients with TNBC who underwent surgical resection at the Department of Surgery, Kansai Medical University Hospital (Osaka, Japan), between January 2006 and December 2018. Patients administered neoadjuvant chemotherapy or those with a special type of invasive carcinoma were excluded, because chemotherapy may influence the expression of PD-L1 and/or CD155, and may affect prognosis [20]. Eventually, 61 patients with TNBC were included in this study. The patient cohort was fundamentally the same as that of our previous study [20], in which we analyzed the relationship between adipophilin expression, a lipid droplet-associated protein, and the clinicopathological features of patients with TNBC. The contents of this paper, analyzing the significance of CD155, an immune checkpoint protein, in patients with TNBC, do not overlap with those of our previous study [20].

This retrospective single-center study was in accordance with the ethical standards of the institutional and/or national research committee and with the 1964 Helsinki Declaration and

its later amendments or comparable ethical standards. The study protocol was approved by the Institutional Review Board of Kansai Medical University Hospital (approval number: #2019041). Informed consent was obtained from all patients by the opt-out method, which was approved by the Institutional Review Board of Kansai Medical University Hospital (approval number: #2019041) owing to the retrospective nature of the study, with no risk to the participants. Information regarding the study, such as the inclusion criteria and opportunity to opt-out, was provided to patients through the institutional website.

## Histopathological analysis

Surgically resected specimens were formalin fixed, sectioned, and stained with hematoxylin and eosin. All histopathological diagnoses were independently evaluated by more than two experienced diagnostic pathologists using the tumor–node–metastasis classification of malignant tumours (eighth edition). Histopathological grading was based on the Nottingham histological grade [21]. The Ki-67 labeling index (LI) was considered high at ≥40%, according to a meta-analysis of patients with TNBC [22], and given that the median Ki-67 LI of this cohort was 40%. Stromal tumor-infiltrating lymphocytes (TILs) were identified using hematoxylin and eosin staining, and were considered high at ≥60% and low/intermediate at <59%, according to the TIL Working Group recommendations [23,24].

## Tissue microarray

Hematoxylin and eosin-stained slides were used to select the most morphologically representative carcinoma regions, and three tissue cores of 2 mm in diameter were punched out from the paraffin-embedded blocks for each patient. Tissue cores were arrayed in the recipient paraffin blocks.

## Immunohistochemistry

Immunohistochemical analyses were performed using an autostainer (Discovery Ultra System; Roche Diagnostics, Basel, Switzerland) according to the manufacturer's instructions. Primary rabbit monoclonal antibody against CD155 (#81254: Cell Signaling Technology; Danvers, MA, USA) was used. The staining intensity of CD155 was classified into four categories as follows: 0, no positive staining of tumor cells; +1, weakly stained tumor cells; +2, moderately stained tumor cells; and +3, strongly stained tumor cells. The number (%) of positive neoplastic cells was counted. Accordingly, the CD155 expression score (range, 0–300) was calculated for each case following the same method of evaluating PD-L1 expression [25]: CD155 expression score = staining intensity × percentage of positive cells. Patients with CD155 expression scores of ≥ 50 were defined as CD155-positive in this study. PD-L1 expression was also analyzed using an autostainer (SP142 assay using a Discovery Ultra System; 73–10 assay using the Leica Bond-III; Leica Biosystems, Bannockburn, IL, USA) according to the manufacturer's instructions. Primary rabbit monoclonal antibodies (SP142 assay, Roche Diagnostic; 73–10 assay, Leica Biosystems) were used to detect PD-L1 expression. At least two researchers independently evaluated the immunohistochemical staining results.

## Statistical analysis

All statistical analyses were performed using SPSS Statistics (version 25.0; IBM, Armonk, NY, USA). Correlations between two groups were calculated using the Fisher's exact test for categorical variables and the Mann–Whitney $U$ test for continuous variables. Relapse-free survival (RFS) and overall survival (OS) rates were evaluated using the Kaplan–Meier method, and log-

rank tests were used to compare the groups. A p-value <0.05 was considered statistically significant.

## Results

### Patients' characteristics

This study comprised 61 patients. The clinical characteristics are summarized in Table 1. The median age at initial diagnosis was 67 (range, 31–93) years. All patients were diagnosed with TNBC based on biopsy results, and all samples were of invasive carcinoma, no special type. The median tumor diameter was 20 (range, 2–55) mm. Patients were staged as I (25 patients), IIA (23 patients), IIB (five patients), IIIA (four patients), IIIB (three patients), and IIIC (one patient). Based on the histopathology, two, 27, and 32 patients exhibited grades 1, 2, and 3, respectively. The Ki-67 LI was high, low, and not tested in 36, 21, and four patients, respectively. For those not tested, resected specimens were not available for evaluating the Ki-67 LI, and the expression of hormone receptors and HER2 protein, due to the relatively long postoperative period. In this study, lymph node status was not evaluated in 14 patients (23.0%), because these patients were clinically lymph node-negative and/or refused sentinel lymph node biopsy. None of the patients had lymph node recurrence. The median observation time was 60 (range, 11–163) months. Ten patients (16.4%) experienced relapse (all had distant metastases and none had local recurrence). Nine patients (14.8%) died from the disease; five patients (8.2%) died from other causes.

### CD155 expression and clinicopathological factors

Twenty-five of the 61 patients (41.0%) were CD155-positive (Fig 1A), while the remaining 36 patients (59.0%) were CD155-negative (Fig 1B). Table 2 summarizes the correlation between CD155 expression and clinicopathological factors in the present cohort. CD155 expression did not correlate with any of the clinical factors, including age and the administration of adjuvant chemotherapy. CD155 expression also did not correlate with any of the pathological factors, including pathological stage, Nottingham histological grade, lymphovascular invasion, Ki-67 LI, and stromal TILs.

### Correlation between CD155 and PD-L1 expression

Table 3 shows the correlation between CD155 and PD-L1 expression analyzed using 73–10 and SP142 assays. Although CD155 expression did not correlate with PD-L1 expression in tumor cells in the 73–10 assay (p = 0.115), PD-L1 expression in tumor cells using the SP142 assay was significantly correlated with CD155 expression (p = 0.035). In the 73–10 assay, 24 of 61 patients (39.3%) had PD-L1-positive tumor cells, while 36 of 61 patients (59.0%) demonstrated CD155 and/or PD-L1 expression in tumor cells. Moreover, using the SP142 assay, 25 of 60 patients (41.7%) had PD-L1-positive immune cells, while 38 of 60 patients (63.3%) demonstrated CD155 and/or PD-L1 expression in immune cells (Table 3).

### Correlation between CD155 expression and survival

Fig 2 shows the RFS and OS curves of CD155-positive and -negative patients with TNBC. CD155 expression did not correlate with either RFS or OS (p = 0.485 and 0.843, respectively).

## Discussion

In this study, CD155 expression was observed in 41.0% of patients with TNBC. Its expression did not correlate with any of the clinicopathological factors. Moreover, CD155 expression did

**Table 1. Clinical characteristics of patients with triple-negative breast cancer.**

| Characteristic | Patients (n = 61) |
|---|---|
| Age (years), median (range) | 67 (31–93) |
| Menopausal status, n (%) | |
| Premenopausal | 9 (14.8) |
| Postmenopausal | 51 (83.6) |
| Unknown | 1 (1.6) |
| Tumor size (mm), median (range) | 20 (2–55) |
| Pathological stage, n (%) | |
| I | 25 (41.0) |
| IIA | 23 (37.7) |
| IIB | 5 (8.2) |
| IIIA | 4 (6.6) |
| IIIB | 3 (4.9) |
| cIIIC | 1 (1.6) |
| Lymph node status, n (%) | |
| Positive | 14 (23.0) |
| Negative | 33 (54.0) |
| Not tested | 14 (23.0) |
| Lymphatic invasion, n (%) | |
| Positive | 52 (85.2) |
| Negative | 9 (14.8) |
| Venous invasion, n (%) | |
| Positive | 37 (60.7) |
| Negative | 24 (39.3) |
| Nottingham histological grade, n (%) | |
| 1 | 2 (3.3) |
| 2 | 27 (44.3) |
| 3 | 32 (52.5) |
| Ki-67 LI, n (%) | |
| High | 36 (59.0) |
| Low | 21 (34.4) |
| Not tested | 4 (6.6) |
| Stromal TILs, n (%) | |
| LPBC | 19 (31.1) |
| Non-LPBC | 42 (68.9) |
| PD-L1 expression in immune cells[a] | |
| Positive | 37 (60.7) |
| Negative | 24 (39.3) |
| Adjuvant chemotherapy, n (%) | |
| Administered | 34 (55.7) |
| Not administered | 24 (39.3) |
| Unknown | 3 (4.9) |

[a] 73–10 assay.

LI, labeling index; LPBC, lymphocyte-predominant breast cancer; PD-L1, programmed death ligand 1; TIL, tumor-infiltrating lymphocyte.

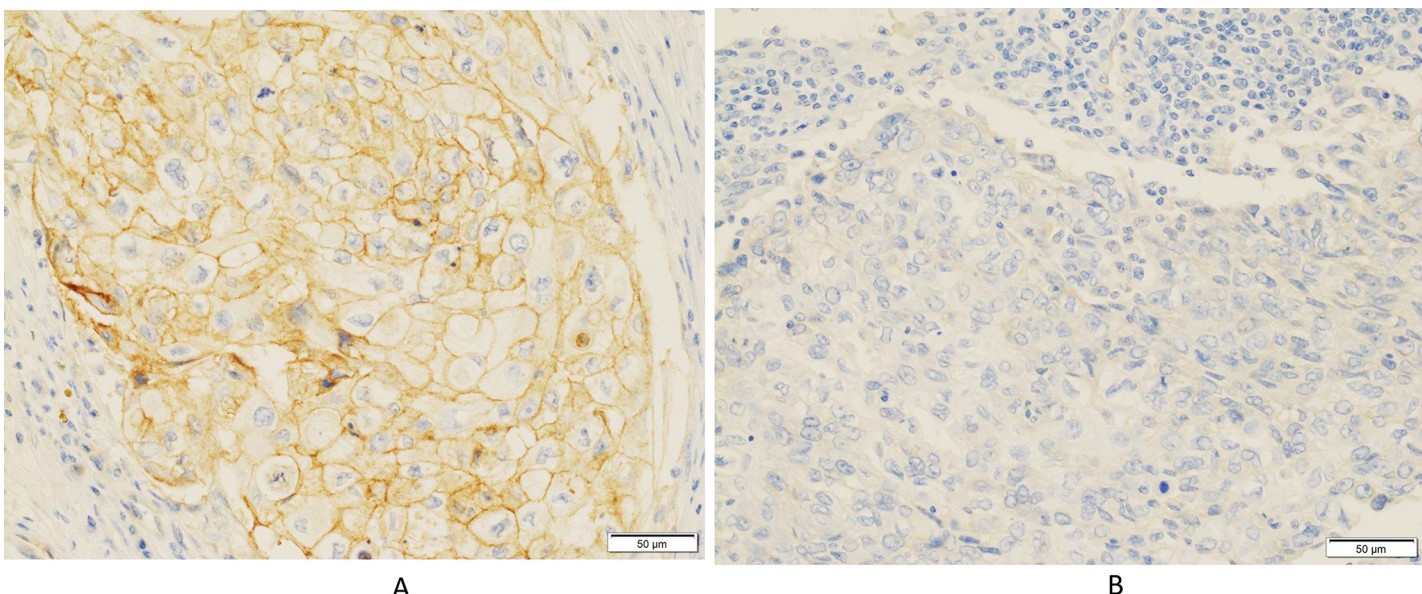

**Fig 1. Immunohistochemical Staining for CD155 in patients with triple-negative breast cancer.** (**A**) Complete and strong membranous immunoreactivity in tumor cells (× 400). (**B**) No immunoreactivity in tumor cells (× 400).

not correlate with PD-L1 expression in tumor cells using the 73–10 assay, but was significantly correlated with PD-L1 expression in tumor cells using the SP142 assay, with 59.0% of patients demonstrating CD155 and/or PD-L1 expression in tumor cells using the 73–10 assay, and 63.3% of patients demonstrating CD155 and/or PD-L1 expression in immune cells using the SP142 assay.

There have been few reports addressing CD155 expression in breast cancer; thus, the clinicopathological features and prognostic significance of CD155 expression have not been established [17–19]. The rate of CD155 expression in patients with breast cancer (all subtypes) ranged from 38.1% (48 of 126 patients) [17] to 52.3% (113 of 216 patients) [19]. Overexpression of CD155 was significantly higher in TNBC than in luminal A type breast cancer in one study [17]; however, this tendency was not observed in another study [19]. Patients with CD155 overexpression had a significantly higher Ki-67 LI [17,19], abundant TILs [17,19], and the presence of infiltrating NK cells [18] and macrophages [19]. CD155 was also significantly associated with a higher tumor stage and the presence of lymph node metastasis in the report by Yong *et al.* [19], although this association was not confirmed in another study by Li *et al.* [17]. The prognostic significance of CD155 expression has not yet been established given that CD155 expression was shown to be a significant poor prognostic factor in two studies [17,19], but a favorable prognostic factor in another study [18]. These studies included all subtypes of breast cancer. However, no study has focused on the relationship between CD155 expression and the clinicopathological features of patients with TNBC. Moreover, the association between CD155 and PD-L1 expression has not yet been studied. Herein, we analyzed the relationship between CD155 expression and the clinicopathological features of patients with TNBC. CD155 expression was not a significant factor for RFS and OS, and did not correlate with pathological stage, lymph node status, the Ki-67 LI, or stromal TILs in this cohort, although the proportion of CD155-positive patients (41.0%) was consistent with those of previous reports [17,19]. Thus, the significance of CD155 expression may differ among the various molecular subtypes of breast cancer.

**Table 2. Correlation between clinicopathological factors and CD155 expression.**

| Factors | CD155-Positive (n = 25) | CD155-Negative (n = 36) | p-Value |
|---|---|---|---|
| **Age (years), median ± SD** | 64 ± 17 | 67 ± 13 | 0.567 |
| **Menopausal status, n** | | | |
| **Premenopausal** | 6 | 3 | 0.145 |
| **Postmenopausal** | 19 | 32 | |
| **Unknown** | 0 | 1 | |
| **Tumor size (mm), n** | | | |
| **≤ 20** | 14 | 19 | 1.000 |
| **> 20** | 11 | 17 | |
| **Pathological stage, n** | | | |
| **I–II** | 23 | 30 | 0.452 |
| **III** | 2 | 6 | |
| **Lymph node status, n** | | | |
| **Positive** | 6 | 8 | 1.000 |
| **Negative** | 15 | 18 | |
| **Not tested** | 4 | 10 | |
| **Lymphatic invasion, n** | | | |
| **Positive** | 21 | 31 | 1.000 |
| **Negative** | 4 | 5 | |
| **Venous invasion, n** | | | |
| **Positive** | 13 | 24 | 0.294 |
| **Negative** | 12 | 12 | |
| **Nottingham histological grade, n** | | | |
| **1–2** | 11 | 18 | 0.795 |
| **3** | 14 | 18 | |
| **Ki-67 LI, n** | | | |
| **High** | 18 | 18 | 0.166 |
| **Low** | 6 | 15 | |
| **Not tested** | 1 | 3 | |
| **Stromal TILs, n** | | | |
| **LPBC** | 10 | 9 | 0.793 |
| **Non-LPBC** | 15 | 27 | |
| **Adjuvant chemotherapy, n** | | | |
| | 12 | 22 | 0.291 |
| | 12 | 12 | |
| | 1 | 2 | |

LI, labeling index; LPBC, lymphocyte-predominant breast cancer; SD, standard deviation; TIL, tumor-infiltrating lymphocyte.

Furthermore, this study analyzed the relationship between CD155 and PD-L1 expression in TNBC for the first time. Several immunohistochemical assays have been independently developed as companion diagnostics to determine the indications for anti-PD-L1 therapy, and differences in positive immunoreactivity among primary PD-L1 antibodies are well known [26,27]. We evaluated PD-L1 expression using SP142 and 73–10 assays. The SP142 assay has been developed as a companion diagnostic for anti-PD-L1 immunotherapy in breast cancer. It targets a different binding site than the 73–10 assay. The SP142 assay targets the C-terminal cytoplasmic domain of PD-L1, while the 73–10 assay targets the intracytoplasmic domain [26,28]. Differences in positive immunoreactivity for PD-L1 in cancer-associated fibroblasts of

**Table 3. Correlation between CD155 and PD-L1 expression.**

| PD-L1 Expression | CD155-Positive (n = 25) | CD155-Negative (n = 36) | p-Value |
|---|---|---|---|
| **73–10 assay** | | | |
| Tumor cells, n | | | 0.115 |
| Positive | 13 | 11 | |
| Negative | 12 | 25 | |
| SP-142 assay | | | |
| Tumor cells, n | | | **0.035** |
| Positive | 10 | 5 | |
| Negative | 15 | 30 | |
| Not tested | 0 | 1 | |
| Immune cells, n | | | 0.437 |
| Positive | 12 | 13 | |
| Negative | 13 | 22 | |
| Not tested | 0 | 1 | |

PD-L1, programmed death ligand 1.

TNBC tissues has been reported [28]. In this study, CD155 expression was significantly corre-lated with PD-L1 expression in tumor cells using the SP142 assay, but not the 73–10 assay. This may be due to differences in the binding sites of the primary PD-L1 antibodies.

Although anti-PD-L1-targeted therapy significantly improves the survival of PD-L1-posi-tive patients with TNBC [16], the number of patients with breast cancer who can benefit from anti-PD-L1-targeted therapy is small. Thus, novel targets for anti-cancer immunotherapy are needed to improve the prognosis of patients with TNBC. It has been speculated that CD155 expression leads to tumor immunosuppression [9], because the interaction of CD155 with TIGIT or CD96-positive T lymphocytes and NK cells results in exhaustion of immune

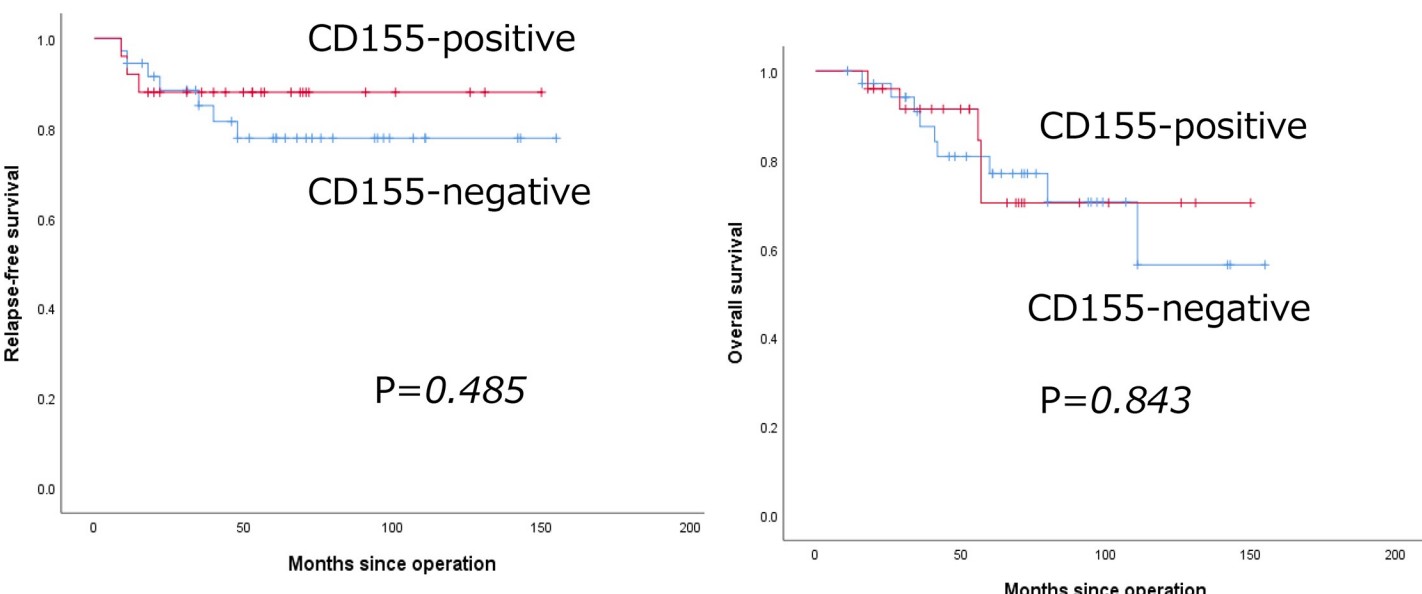

**Fig 2. Kaplan–Meier curves for relapse-free survival and overall survival in patients with triple-negative breast cancer.** Left: Relapse-free survival curves of CD155-positive (red line) and -negative (blue line) patients. Right: Overall survival curves of CD155-positive (red line) and -negative (blue line) patients.

function, in addition to reducing the secretion of interferon-γ [29,30]. Thus, blockade of CD155–TIGIT or CD96 signaling could improve antitumor immune cell function. Clinical trials targeting TIGIT are ongoing [9]. A recent study [31] demonstrated that CD155 expression in tumor cells was associated with resistance to anti-PD-1 immunotherapy in patients with metastatic malignant melanoma, and it has been speculated that blockade of CD155 may improve the response to anti-PD-1 therapy. Therefore, CD155-positive patients with TNBC may be candidates for further immunotherapy. The results of this study indicate that CD155 expression may expand the pool of candidates with TNBC who could benefit from antitumor immunotherapy, because of the number of patients with TNBC demonstrating CD155 and/or PD-L1 expression using 73–10 and SP142 assays, in comparison to that for PD-L1 expression. Thus, analysis of CD155 expression, as well as PD-L1 expression, may be useful for cancer immunotherapy.

It is important to note the limitations of this study. First, this was a retrospective single-center study with a relatively small sample size, which could have led to selection bias. Second, we used tissue microarray cores to analyze the immunostaining for CD155 and PD-L1, similar to a previously reported method for CD155 expression [19]. There could be heterogeneous expression in the cancer tissue, although we selected regions that were most morphologically representative of the cancer tissue. Therefore, further studies with larger sample sizes are required to validate our results. Furthermore, chemotherapy may affect CD155 expression status in TNBC [32]; consequently, this study excluded patients who had received neoadjuvant chemotherapy. Additional studies are needed to clarify the expression status of CD155 in patients with TNBC, with and without neoadjuvant chemotherapy.

## Conclusions

This study demonstrated that CD155 expression did not significantly correlate with clinicopathological factors or prognosis in patients with TNBC. CD155 expression was significantly correlated with PD-L1 expression in tumor cells using the SP142 assay, but not the 73–10 assay. In the 73–10 assay, 59% of patients with TNBC demonstrated CD155 and/or PD-L1 expression in tumor cells, while 39.3% of patients demonstrated only PD-L1 expression in tumor cells. In the SP142 assay, 63.3% of patients with TNBC demonstrated CD155 and/or PD-L1 expression in immune cells. Our findings indicate that CD155 expression may expand the pool of candidates with TNBC who could benefit from antitumor immunotherapy.

## Supporting information

**S1 Table. Clinicopathological characteristics and expression profiles of PD-L1 and CD155.** (PDF)

## Author Contributions

**Conceptualization:** Katsuhiro Yoshikawa.

**Data curation:** Katsuhiro Yoshikawa.

**Formal analysis:** Katsuhiro Yoshikawa.

**Investigation:** Katsuhiro Yoshikawa.

**Methodology:** Katsuhiro Yoshikawa.

**Resources:** Katsuhiro Yoshikawa.

**Supervision:** Mitsuaki Ishida, Hirotsugu Yanai.

**Visualization:** Katsuhiro Yoshikawa.

**Writing – original draft:** Katsuhiro Yoshikawa.

**Writing – review & editing:** Katsuhiro Yoshikawa, Mitsuaki Ishida, Hirotsugu Yanai, Koji Tsuta, Mitsugu Sekimoto, Tomoharu Sugie.

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
