## [Decision Letter · Decision Letter 0]

15 Mar 2021

PONE-D-21-03264

Immunohistochemical analysis of CD155 expression in triple-negative breast cancer patients

PLOS ONE

Dear Dr. Mitsuaki Ishida,

Thank you for submitting your manuscript to PLOS ONE. After careful consideration, we feel that it has merit but does not fully meet PLOS ONE’s publication criteria as it currently stands. Therefore, we invite you to submit a revised version of the manuscript that addresses the points raised during the review process.

We look forward to receiving your revised manuscript.

Kind regards,

Ramon Andrade De Mello, MD, PhD, FACP

Academic Editor

PLOS ONE

Journal Requirements:

2. Please state whether the ethics committee that approved your study also approved the opt-out consent procedure.

3. In the ethics statement in the manuscript and in the online submission form, please provide additional information about the patient records/samples used in your retrospective study, including: a) whether all data were fully anonymized before you accessed them; b) the date range (month and year) during which patients' medical records/samples were accessed.

4.Thank you for stating the following financial disclosure:

 "No."

Reviewers' comments:

Reviewer's Responses to Questions

**Comments to the Author**

1. Is the manuscript technically sound, and do the data support the conclusions?

Reviewer #1: Partly

Reviewer #2: Partly

2. Has the statistical analysis been performed appropriately and rigorously? 

Reviewer #1: Yes

Reviewer #2: N/A

3. Have the authors made all data underlying the findings in their manuscript fully available?

Reviewer #1: Yes

Reviewer #2: No

4. Is the manuscript presented in an intelligible fashion and written in standard English?

Reviewer #1: Yes

Reviewer #2: Yes

5. Review Comments to the Author

Reviewer #1: To the critical analysis of the article, I suggest the revision of the following topics:

- In line 86, could the authors describe why patients undergoing neoadjuvant chemotherapy were not included?

- In line 107, the study by Wu Q et al is cited as a reference for a high Ki 67 index. However, this meta-analysis was unable to adequately perform the subgroup analysis, due to the heterogeneity of the results. Thus, I suggest revising the inclusion of this data.

- In line 127 the calculation of aCD 155 expression score is mentioned, but there is no support for this information. I suggest the discrimination on the validity of that score and inclusion of the bibliographic reference.

- In line 130, explain why the use of two methods for analyzing the PD-L1 expression, and what basic difference between them, since the results between the methods were not comparable.

- What is the external validity of the study if the average size of the evaluated tumors was 20 mm and according to the National Comprehensive Cancer Network (NCCN) triple negative breast cancer guideline, tumors larger than 10 mm are indicated for the initiation of therapy with neoadjuvant chemotherapy?

- Explain why 36 patients (59%) have a stage higher than IIA and how this could influence the results found?

- Discuss in the text why 14 patients in the cohort (23%) did not undergo an axillary evaluation. Would it not be valid to exclude them from the analysis, since the TNM Classification of Malignant Tumours, 8th Edition-based staging is incomplete and this would not allow a reliable analysis of the data?

- Address in the text the discussion of why four patients (6.6%) did not have a Ki67% assessment.

- Explain why 24 patients in the study (39%) did not undergo adjuvant treatment and how this data relates to the reliability of the oncological outcome of this sample.

- Assess whether the non-relationship between the expression of the biomarker CD 155 and outcomes of mortality and locoregional recurrence (OS, RFS) are not biased by the sample's heterogeneity.

Reviewer #2: FINANCIAL DISCLOSURE NEED TO ADD STATEMENT WITH DETAILS.

PENDING ETHICS STATEMENT PLEASE ADD APPROVAL NUMBER (IF PRESENT) AND FORM OF CONSENT OBTAINED.

WHY DATA IS NOT AVALIABLE? DESCRIBE WHERE AND WHY THEY MAY NOT BE FOUND

I BELIEVE THE CRITERIA FOR PREDICTION OF OUTCOME COULD BE BETTER EXPLORED, FOR EXEMPLE: DOES THE PACIENTS WITH POOR PERFORMANCE STATUS NEEDED TO USE PROLONGED ANTIBIOTICS? PDL1 COULD BE A VALUABE DATA IN THE STUDY FOR BETTER RESULTS TOO. RESUME GRAPHICS, NO NEED FOR ALL OF THEM.

6. PLOS authors have the option to publish the peer review history of their article (what does this mean?). If published, this will include your full peer review and any attached files.

Reviewer #1: No

Reviewer #2: No

---

## [Author Response · Author response to Decision Letter 0]

20 Apr 2021

April 20, 2021

Joerg Heber

Editor-in-Chief

PLOS ONE

Dear Editor:

We would like to resubmit the manuscript titled “Immunohistochemical analysis of CD155 expression in triple-negative breast cancer patients.” The manuscript ID is PONE-D-21-03264.

We thank the Editor and the reviewers for their thoughtful suggestions and insights. The manuscript has benefited from these insightful suggestions. We look forward to working with the Editor to move this manuscript closer to publication in PLOS ONE.

The manuscript has been rechecked and the necessary changes have been made (these revisions are highlighted in red font in the edited file) in accordance with the reviewers’ suggestions. The responses to all comments have been prepared and are attached herewith.

According to the Editor’s suggestions, we delated the ethics statement in the Declarations section. We added the statements of Funding and Competing interests in the Declarations section. Moreover, we added the comment regarding the opt-out consent in the Materials and Methods (page 6-7). 

Thank you for your continued consideration. We look forward to hearing from you.

Sincerely,

Mitsuaki Ishida, M.D., Ph.D.

Department of Pathology and Clinical Laboratory, Kansai Medical University, 2-5-1 Shinmachi, Hirakata City, Osaka 573-1010, Japan

Tel.: +81-72-804-2794

Fax: +81-72-804-2794

Email: ishidamt@hirakata.kmu.ac.jp

Reviewer #1

Thank you very much for reviewing our manuscript. We appreciate your constructive comments. We have made the following revisions in response to the issues you raised.

Comment 1: In line 86, could the authors describe why patients undergoing neoadjuvant chemotherapy were not included?

Response: Thank you for your comment. We have revised this as follows: “Patients administered neoadjuvant chemotherapy or those with a special type of invasive carcinoma were excluded, because chemotherapy may influence the expression of PD-L1 and/or CD155, and may affect prognosis [20].” (page 6, lines 78–81)

Comment 2: In line 107, the study by Wu Q et al is cited as a reference for a high Ki 67 index. However, this meta-analysis was unable to adequately perform the subgroup analysis, due to the heterogeneity of the results. Thus, I suggest revising the inclusion of this data.

Response: Thank you for your comment. As you surmised, the results of the meta-analysis by Wu et al. are controversial. Even though the Ki-67 LI is widely used, its reproducibility is not high. Few meta-analyses have examined the optimal cut-off value of the Ki-67 LI for TNBC. Thus, we cited the Wu et al. study in this paper. Additionally, we determined the median Ki-67 LI to be 40% in the present cohort of patients with TNBC. Therefore, we set the cut-off value of the Ki-67 LI at 40%. We have revised the Materials and methods section as follows: “The Ki-67 labeling index (LI) was considered high at ≥40%, according to a meta-analysis of patients with TNBC [22], and given that the median Ki-67 LI of this cohort was 40%.” (page 7, lines 101–103)

Comment 3: In line 127 the calculation of aCD 155 expression score is mentioned, but there is no support for this information. I suggest the discrimination on the validity of that score and inclusion of the bibliographic reference.

Response: Thank you for your comment. We have revised this as follows: “Accordingly, the CD155 expression score (range, 0–300) was calculated for each case following the same method of evaluating PD-L1 expression [25].” (page 8, lines 120–121)

Comment 4: In line 130, explain why the use of two methods for analyzing the PD-L1 expression, and what basic difference between them, since the results between the methods were not comparable.

Response: Thank you for your comment. We have added the following explanation to the Discussion section: “We evaluated PD-L1 expression using SP142 and 73–10 assays. The SP142 assay has been developed as a companion diagnostic for anti-PD-L1 immunotherapy in breast cancer. It targets a different binding site than the 73–10 assay. The SP142 assay targets the C-terminal cytoplasmic domain of PD-L1, while the 73–10 assay targets the intracytoplasmic domain [26, 28]. Differences in positive immunoreactivity for PD-L1 in cancer associated-fibroblasts of TNBC tissues has been reported [28].” (page 21, lines 237–243)

Comment 5: What is the external validity of the study if the average size of the evaluated tumors was 20 mm and according to the National Comprehensive Cancer Network (NCCN) triple negative breast cancer guideline, tumors larger than 10 mm are indicated for the initiation of therapy with neoadjuvant chemotherapy?

Response: Thank you for your comment. The National Comprehensive Cancer Network guidelines do recommend that TNBC patients with tumors larger than 10 mm are indicated for neoadjuvant chemotherapy. The present cohort included TNBC patients with tumors larger than 10 mm who refused neoadjuvant chemotherapy. Tumors 20 mm or smaller are classified as T1, while those larger than 20 mm are classified as T2. The median tumor size of the present cohort was 20 mm. Thus, we compared CD155 expression status and tumor size using a cut-off value of 20 mm.

Comment 6: Explain why 36 patients (59%) have a stage higher than IIA and how this could influence the results found?

Response: Thank you for your comment. The present cohort included patients with stage I (41%) and stage II–III (59%) TNBC. According to the most recent national survey performed by Japanese Breast Cancer Society, 57.3–59.9% of patients with TNBC were staged higher than IIA, consistent with our cohort. 

Comment 7: Discuss in the text why 14 patients in the cohort (23%) did not undergo an axillary evaluation. Would it not be valid to exclude them from the analysis, since the TNM Classification of Malignant Tumours, 8th Edition-based staging is incomplete and this would not allow a reliable analysis of the data?

Response: Thank you for your comment. We have added the following sentences to the Materials and methods section: “In this study, lymph node status was not evaluated in 14 patients (23.0%), because these patients were clinically lymph node-negative and/or refused sentinel lymph node biopsy. None of the patients had lymph node recurrence.” (page 10, lines 148–151)

Comment 8: Address in the text the discussion of why four patients (6.6%) did not have a Ki67% assessment.

Response: Thank you for your comment. We have added the following sentences to the Materials and methods section: “For those not tested, resected specimens were not available for evaluating the Ki-67 LI, and the expression of hormone receptors and HER2 protein, due to the relatively long postoperative period.” (page 10, lines 146–148)

Comment 9: Explain why 24 patients in the study (39%) did not undergo adjuvant treatment and how this data relates to the reliability of the oncological outcome of this sample.

Response: Thank you for your comment. In this study, 24 patients chose not to receive adjuvant chemotherapy. CD155 expression did not correlate with the administration of adjuvant chemotherapy, although there is the potential for this to affect outcomes.

Comment 10: Assess whether the non-relationship between the expression of the biomarker CD 155 and outcomes of mortality and locoregional recurrence (OS, RFS) are not biased by the sample's heterogeneity.

Response: Thank you for your comment. CD155 expression did not significantly correlate with prognosis in patients with TNBC in the present cohort. As discussed, this study had a relatively small sample size, which could have led to selection bias. Sample heterogeneity may have also influenced the results of this study. Therefore, further studies with larger sample sizes are required to validate our findings. 

Reviewer #2

Thank you very much for reviewing our manuscript. We appreciate your constructive comments. We have made the following revisions in response to the issue you raised.

Comment: Very interesting article, patients were well selected and study were well conducted. The statistical analysis are fine. The only suggestion is in the discussion instead of only using the number of the reference in the text like "in one report [19], although this association was not confirmed in another study [17]" I would also use the authors names to easy comprehension.

Response: Thank you for your kind comment. As you suggested, we changed the sentence in the Discussion section: “CD155 was also significantly associated with a higher tumor stage and the presence of lymph node metastasis in the report by Yong et al. [19], although this association was not confirmed in another study by Li et al. [17].” (page 20 line 218-220)

---

## [Editor Report · Decision Letter 1]

31 May 2021

Immunohistochemical analysis of CD155 expression in triple-negative breast cancer patients

PONE-D-21-03264R1

Dear Dr. Mitsuaki Ishida,

We’re pleased to inform you that your manuscript has been judged scientifically suitable for publication and will be formally accepted for publication once it meets all outstanding technical requirements.

Kind regards,

Ramon Andrade De Mello, MD, PhD, FACP

Academic Editor

PLOS ONE

---

## [Editor Report · Acceptance letter]

4 Jun 2021

PONE-D-21-03264R1 

Immunohistochemical analysis of CD155 expression in triple-negative breast cancer patients 

Dear Dr. Ishida:

I'm pleased to inform you that your manuscript has been deemed suitable for publication in PLOS ONE. Congratulations! Your manuscript is now with our production department. 

Kind regards, 

on behalf of

Professor Ramon Andrade De Mello 

Academic Editor

PLOS ONE